UPDATE ARTICLE

# Cep131-Cep162 and Cby-Fam92 complexes cooperatively maintain Cep290 at the basal body and contribute to ciliogenesis initiation

Zhimao Wu[1]ᵒ, Huicheng Chen[1]ᵒ, Yingying Zhang[1]ᵒ, Yaru Wang[1,2], Qiaoling Wang[3], Céline Augière[4], Yanan Hou[1], Yuejun Fu[2], Ying Peng[3], Bénédicte Durand [4]*, Qing Wei [1,5,6]*

1 Center for Energy Metabolism and Reproduction, Institute of Biomedicine and Biotechnology, Shenzhen Institutes of Advanced Technology, Chinese Academy of Sciences (CAS), Shenzhen, China, 2 Key Laboratory of Chemical Biology and Molecular Engineering of Ministry of Education, Institute of Biotechnology, Shanxi University, Taiyuan, China, 3 Institute of Medicine and Pharmaceutical Sciences, Zhengzhou University, Zhengzhou, China, 4 University Claude Bernard Lyon 1, MeLiS—UCBL—CNRS UMR 5284—INSERM U1314, Lyon, France, 5 Shenzhen Key Laboratory of Metabolic Health, Shenzhen, China, 6 School of Synthetic Biology, Shanxi Key Laboratory of Nucleic Acid Biopesticides, Shanxi University, Taiyuan, China

ᵒ These authors contributed equally to this work.
* benedicte.durand@univ-lyon1.fr (BD); qing.wei@siat.ac.cn (QW)

The Editors encourage authors to publish research updates to this article type. Please follow the link in the citation below to view any related articles.

## Abstract

Cilia play critical roles in cell signal transduction and organ development. Defects in cilia function result in a variety of genetic disorders. Cep290 is an evolutionarily conserved ciliopathy protein that bridges the ciliary membrane and axoneme at the basal body (BB) and plays critical roles in the initiation of ciliogenesis and TZ assembly. How Cep290 is maintained at BB and whether axonemal and ciliary membrane localized cues converge to determine the localization of Cep290 remain unknown. Here, we report that the Cep131-Cep162 module near the axoneme and the Cby-Fam92 module close to the membrane synergistically control the BB localization of Cep290 and the subsequent initiation of ciliogenesis in *Drosophila*. Concurrent deletion of any protein of the Cep131-Cep162 module and of the Cby-Fam92 module leads to a complete loss of Cep290 from BB and blocks ciliogenesis at its initiation stage. Our results reveal that the first step of ciliogenesis strictly depends on cooperative and retroactive interactions between Cep131-Cep162, Cby-Fam92 and Cep290, which may contribute to the complex pathogenesis of Cep290-related ciliopathies.

## Introduction

Cilia are microtubule-based organelles that extend from the surface of many cell types and are widely present in eukaryotes. They play crucial roles in the development and maintenance of various organs in humans [1–4], and their dysfunction has been linked to a wide range of human genetics diseases known as ciliopathies [5–7].

The structure of cilia remains highly conserved across evolution [8]. A cilium comprises a basal body (BB), a transition zone (TZ), an axoneme and its overlying membrane. The BB

**Data Availability Statement:** All relevant data are within the paper and its Supporting Information files.

**Funding:** National Natural Science Foundation of China (Grant 32070692) to Q.W.; China Postdoctoral Science Foundation (Grant 2022M713288) and National Natural Science Foundation Youth Project of China (Grant 32300564) to Z. W.; Agence Nationale de la Recherche (ANR) grant (ANR-17-CE13-0023-01, Divercil) to B. D.; The funders had no role in study design, data collection and analysis, decision to publish, or preparation of the manuscript.

**Competing interests:** The authors have declared that no competing interests exist.

**Abbreviations:** APF, after puparium formation; BB, basal body; BBS, Bardet–Biedl syndrome; IFT, intraflagellar transport; JBTS, Joubert syndrome; LCA, Leber congenital amaurosis; MKS, Meckel–Gruber syndrome; SLS, Senior–Loken syndrome; TZ, transition zone; WT, wild type.

originates from the mother centriole and attaches to the membrane through transition fibers that come from the distal appendages of the mother centriole [9–11]. The TZ is situated just above transition fibers [12] and is characterized by Y-linker structures that connect the axoneme and ciliary membrane, serving as a diffusion barrier to control ciliary protein entry [9]. The axoneme forms the cilium skeleton and consists of a 9-fold array of doublet microtubules, which are templated from the BB microtubules and surrounded by the ciliary membrane [5,8].

The formation of cilia involves 2 main processes: ciliogenesis initiation and axoneme elongation. The elongation of axoneme relies on the intraflagellar transport (IFT), a evolutionarily conserved transport machinery within cilia [13,14]. The initiation of ciliogenesis involves the membrane docking of BB and the formation of ciliary bud [10,15,16]. In mammals, the membrane docking of BB is mediated by centriole distal appendages/ciliary transition fibers [17–21]. However in invertebrate model organism *Drosophila*, transition fiber proteins are dispensable for BB membrane docking [22], suggesting the presence of alternative mechanisms. The ciliary bud consists of the TZ and its surrounding membrane [10,23,24]. Dozens of proteins have been identified as components of TZ, and mutations in most of them lead to ciliopathy [5]. Studies have categorized TZ proteins into 3 functional modules: Meckel–Gruber syndrome (MKS) module, Nephronophthisis (NPHP) module, and Cep290 [25]. The formation of ciliary membrane relies on the membrane transport regulator small GTPase Rab8 related signaling cascade [26–28]. Dzip1 (DAZ interacting zinc finger protein 1) and its close paralog Dzip1L have recently been found to play a role in ciliary bud formation in both *Drosophila* and mammals [24,29,30]. Downstream of Dzip1/1L, Rab8, and Cby (Chibby)-Fam92 (family with sequence similarity 92) module work together to regulate ciliary membrane formation [24,29,31].

Although speculations have accumulated that ciliary bud formation must be a coordinated event matching the TZ assembly and the ciliary membrane formation [27,29], a detailed molecular linkage between these 2 seemingly independent processes remains largely elusive. Using *Drosophila* model, we provide evidence that the core TZ protein Cep290 play a pivotal role in coordinating ciliary membrane formation and TZ assembly [29]. We demonstrated that, in addition to its classical role in TZ assembly, Cep290 acts upstream of Dzip1, playing an essential role in ciliogenesis initiation and ciliary bud formation [29]. Cep290 is an intriguing cilia gene associated with several ciliopathies [32], including Leber congenital amaurosis (LCA), Senior–Loken syndrome (SLS), Joubert syndrome (JBTS), MKS, and Bardet–Biedl syndrome (BBS). More than 100 ciliopathy mutations have been identified in Cep290 [32]. The broad spectrum of diseases highlights the critical roles of Cep290 in cilia. We and others have demonstrated that the N-terminus of Cep290 associates with the ciliary membrane, while its C-terminus connects with the ciliary axoneme, both being critical for ciliogenesis [29,33,34]. Nonetheless, how Cep290 is targeted to the TZ and whether axoneme derived signal and ciliary membrane localized cues converge to determine the localization and stability of Cep290 remain unknown.

Centrosome protein 131 (Cep131) localizes to both centrosome centriolar satellites and ciliary TZ in mammals [35–37] and has been demonstrated to be required for ciliogenesis, centriole amplification, genome stability, and cancer [35,36,38–40]. Studies on the model organisms *Drosophila* and zebrafish revealed that Cep131 is an evolutionarily conserved BB protein [41,42], and that deletion of Cep131 results in abnormal cilia formation, suggesting that Cep131 has an evolutionarily conserved role in ciliogenesis. But the precise mechanism by which Cep131 regulates ciliogenesis remains largely unknown. In *Drosophila*, Cep131 (also called dilatory in flies) has been shown to localize to the lumen of the distal BB and TZ, playing a role in the initiation of ciliogenesis [42,43]. Interestingly, although both Cep131 and Cby single mutants display mild defects in cilium assembly, the initiation of ciliogenesis is completely

blocked and Cep290 is totally absent from basal bodies in *cep131; cby* double mutant [43]. However, the molecular function of Cep131 in the initiation of ciliogenesis is still largely unknown, and the underlying mechanism by which Cep131 genetically interacts with Cby to regulate Cep290 localization remains unclear.

Here, we report that Cep131 recruits Cep162 to regulate the TZ localization of Cep290 C-terminus and promote ciliogenesis. We show that Cep162 is a Cep131-interacting protein and acts downstream of Cep131 to mediate the association of Cep290 C-terminus with the axonemal microtubules. In addition, we demonstrate that Cby-Fam92 module regulates the TZ localization of the N-terminus of Cep290. Both modules cooperate to recruit and stabilize Cep290 at the TZ, as combined loss of either module (Cep131-Cep162 or Cby-Fam92) results in complete failure of Cep290 localization to the TZ and ultimately prevents the initiation of ciliogenesis. Our results reveal the crucial molecular function of Cep131 in ciliogenesis and unveil a cooperative and orderly assembly of Cep290 facilitated by Cep131-Cep162 and Cby-Fam92 modules during the initiation of ciliogenesis. Thus, our work defines a central molecular pathway composed of 3 modules: Cep131-Cep162, Cep290, and Dzip1-Cby-Fam92, which cooperatively regulate the initiation of cilium assembly.

## Results

### Cep131 is required for the basal body localization of Cep290 C-terminus

To understand the function of the Cep131 in ciliogenesis, we generated the *cep131*[1] (C terminal deletion) mutant flies using the CRISPR-Cas9 system (S1A–S1C Fig). Similar to reported *cep131* mutant [42], our *cep131*[1] flies exhibited typical symptoms related to ciliary defects, and were severely uncoordinated during walking and flying. Consistent with previous reports that *cep131* single mutant fly has mild defects in the initiation of ciliogenesis [42], we observed that approximately 33.6% of *cep131*[1] spermatocyte centrioles showed abnormal elongation of the microtubules labeled by CG6652::GFP (S1D Fig), a *Drosophila* spermatocyte-specific phenotype associated with defects in BB docking and TZ membrane cap formation. Consistent with this observation, the signal of the ciliogenesis initiation regulators Cep290, Dzip1, and Cby, as well as the TZ marker Mks1 were notably reduced at the basal bodies of spermatocytes of *cep131*[1] mutants, although they were still detectable (Fig 1A).

Cep290 is the most upstream protein known in the initiation of ciliogenesis in *Drosophila* [29,30]. Works in fly and mammalian cells have suggested that Cep290 bridges the ciliary axoneme and the membrane, with its C-terminus associated with microtubule doublets and N-terminus associated with the membrane [29,33,34]. Previously, we have demonstrated that both N-terminal truncation (Cep290-N, aa 1–650) and C-terminal truncation (Cep290-C, aa 1385 to the end at 1978) of *Drosophila* Cep290 are capable of localizing to the TZ, independently of each other [29]. 3D-SIM microscopy revealed that Cep290-C::GFP localizes close to the axoneme, whereas Cep290-N::GFP localizes close to the membrane, displaying a noticeably larger diameter [29]. Given that Cep131 localizes to the lumen of TZ [43], we hypothesized that Cep131 might have a role in regulating the localization of Cep290 C-terminus. As expected, we found that the signal of Cep290-C::GFP signal was nearly completely lost in spermatocyte cilia of *cep131* mutants, whereas the signal intensity of Cep290-N::GFP was similar to that in wild-type (WT) (Fig 1A). Interestingly, we noticed that both the diameter of endogenous Cep290 N-terminus (labeled by anti-Cep290 antibody against aa 292–541 in its N-terminus) and Cep290-N::GFP in *cep131* mutants were significantly smaller than that in WT, suggesting a potential alternation in the conformation of Cep290 in *cep131* mutants (Figs 1A and S1E). Collectively, our results indicate that Cep131 specifically regulates the localization of Cep290 C-terminus in spermatocyte cilia.

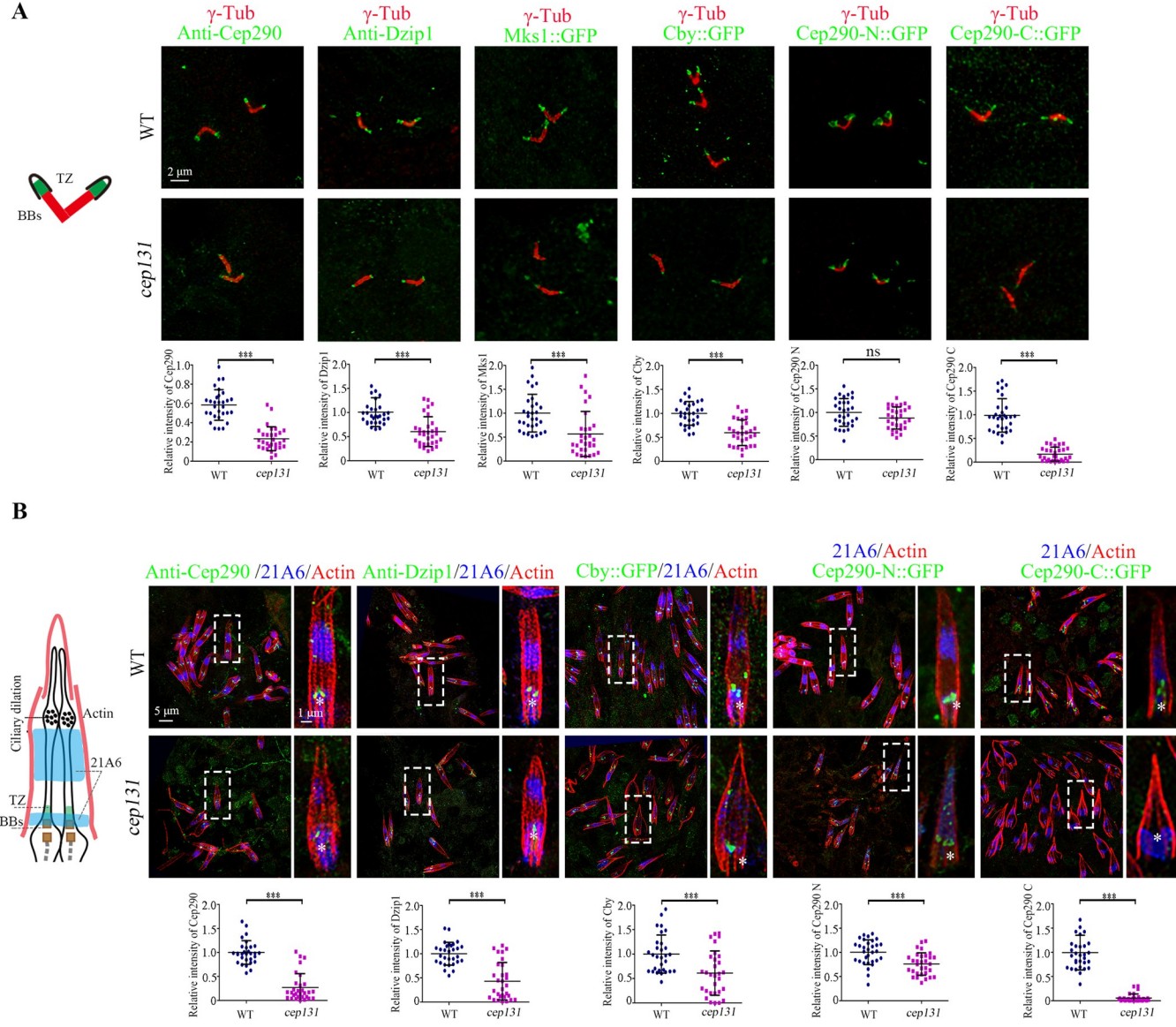

**Fig 1. Cep131 is required for the BB localization of Cep290-C terminus.** (A) Localization of various TZ proteins in spermatocyte cilia of WT flies and *cep131* mutants and quantifications of corresponding relative fluorescence intensities. In *cep131* mutants, the signals of Cep290, Dzip1, Cby::GFP, and Mks1::GFP are significantly reduced compared to WT. Importantly, Cep290-C::GFP signal is almost completely lost in *cep131* mutants. The BB is labeled with γ-Tubulin (red). The error bars represent the mean ± SD, *n* = 30. (B) The localization of various TZ proteins in auditory cilia of WT flies and *cep131* mutants. Similar to what we observed in spermatocyte cilia, the signals of Cep290, Dzip1, and Cby::GFP are significantly decreased, and Cep290-C::GFP is completely lost at the base of the sensory cilia. 21A6 (blue) marks the cilia base, Actin (red) marks the ciliated region. The error bars represent the mean ± SD, *n* = 30. Scale bars: 2 μm (A), 5 μm (B, full-scale images on the left), 1 μm (B, insets or zoomed in areas on the right). The data underlying this figure can be found in S1 Data. BB, basal body; TZ, transition zone; WT, wild type.

To determine whether our observation is specific to spermatocyte cilia, we focused on sensory cilia, another type of cilia in *Drosophila*. Similar to our observation in spermatocytes, the signal intensities of Cep290, Cep290-N, Dzip1, Cby, and Mks1 were mildly affected, whereas Cep290-C::GFP was almost completely lost from the basal bodies in auditory cilia of *cep131* mutants (Fig 1B). Hence, Cep131 also promotes the binding of Cep290 C-terminus to the axoneme in sensory cilia.

## Cep162 bridges Cep131 and Cep290

Next, we wondered whether Cep131 directly interacts with Cep290. However, no interaction between Cep131 and Cep290 was observed in our yeast two-hybrid (Y2H) assay (S2A Fig), suggesting that additional proteins might mediate the functional interaction between Cep131 and Cep290. In mammalian cells, it has been reported that Cep162 localizes to the centriole distal end and interacts with Cep290 to promote its association with microtubules [44]. Protein homology search using NCBI protein–protein BLAST identified CG42699 as the sole homolog of Cep162 in *Drosophila* (S3 Fig). Consistent with reports in mammalian cells, Y2H assay showed that CG42669 interacts with *Drosophila* Cep290 (S2A Fig). Interestingly, our Y2H assay also revealed that CG42699 interacts with *Drosophila* Cep131 (Fig 2A and 2B). Specifically, the C-terminal half of CG42699 (aa 448 to the end at 897) interacts with Cep131, while the N-terminal half does not. GST pull-down assay further confirmed the direct interaction between Cep131 and Cep162 C-terminus, and showed that Cep162 C-terminus interacts with both the N-terminal half (aa 1–549) and C-terminal half (aa 550 to the end at 1114) of Cep131 (S2B Fig). Subsequent analysis using Y2H showed that Cep162 C-terminus does not interact with the middle region of Cep131 (aa 481–781), but it does interacts with both the N-terminus (aa 1–480) and C-terminus (aa 782 to the end at 1114) (S2C Fig), indicating the presence of 2 binding sites for Cep162 in Cep131.

The function of CG42699/Cep162 in fly is unknown yet. We constructed a transgenic fly strain expressing Cep162::GFP under the control of its endogenous promoter to examine its subcellular localization in testis and ciliated sensory neurons. We observed that Cep162 was localized to the BB in all types of cilia (Fig 2C–2E), indicating that the subcellular localization of Cep162 in *Drosophila* is conserved. Interestingly, we found that the BB signal of Cep162 was completely lost in *cep131* mutants (Fig 2C and 2D), indicating that Cep131 plays a critical role in recruiting Cep162. Notably, using transgenic flies expressing a GFP-tagged C-terminal fragment of Cep162 (Cep162-C::GFP) encompassing amino acids 448–897, we observed that Cep162 C-terminus alone was able to target to the BB (Fig 2F). Conversely, the GFP-tagged N-terminal fragment Cep162-N::GFP comprising amino acids 1–447 failed to localize to the BB. These results indicate that the C-terminus of Cep162 is critical for BB targeting. This characteristic is conserved in mammals, as it has been reported previously that the centrosome localization of mammalian Cep162 also depends on its C-terminus [44].

In spermatocyte cilia, Cep162::GFP was localized at the tips of the basal bodies (Fig 2E). To more accurately determine the localization of Cep162, we performed 3D-SIM and examined its spatial distribution with respect to other BB proteins. As shown in Fig 2G, both Cep131::GFP and Cep162::GFP were surrounded by the TZ protein Dzip1. Notably, the plot profile of fluorescence showed that the distribution of Cep162 was bimodal, while the distribution of Cep131 has a single peak. Consistent with this observation, the average radial diameter of Cep162 signal was larger than that formed by Cep131 (Fig 2G and 2H), suggesting that Cep162 surrounds Cep131 (Fig 2I). Furthermore, we calculated the average radial diameters of Cep290-C::GFP and Cep290-N::GFP signals, and found that Cep290-C::GFP was close to Cep162, and formed a smaller diameter domain than that formed by Cep290-N (Fig 2G–2I). Taken together, our 3D-SIM spatial distribution data support the role of Cep162 in mediating the connection between Cep131 and Cep290 C-terminus at the TZ.

In round spermatids, the TZ starts migrating along the growing axoneme. Cep162::GFP also migrated away from the BB with the ring centriole labeled by γ-tubulin, but its signal was gradually decreased and eventually completely disappeared from the ciliary cap base (labeled by Fbf1) in elongating spermatids (Fig 2E). This behavior is similar to that of Cep131 but different from other TZ proteins [43]. Such unique temporal localization pattern of Cep131 and

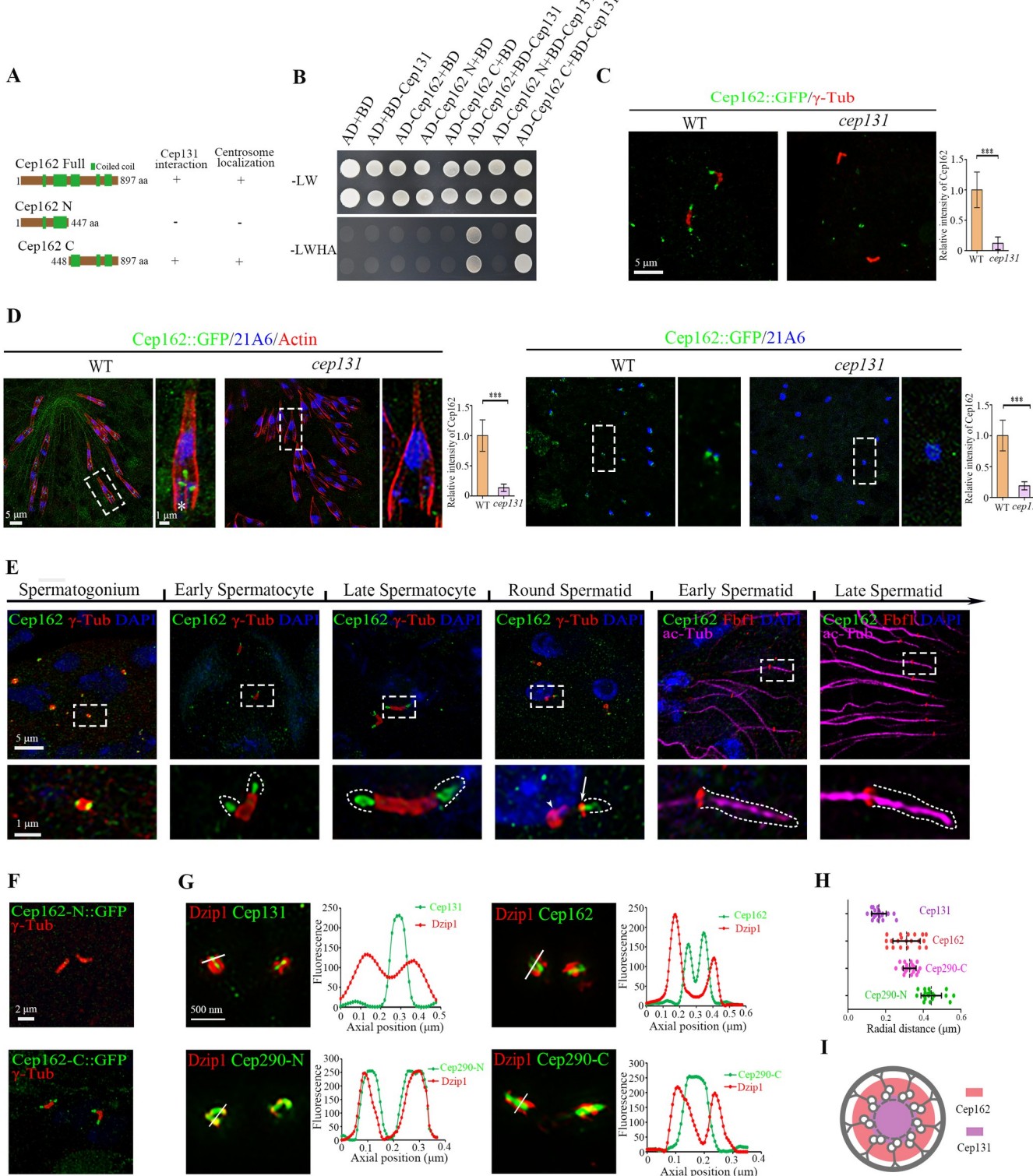

**Fig 2. Cep131 interacts with and recruits Cep162 to centriole tips and the TZ.** (A, B) Cep131 directly interacts with Cep162. (A) Schematic representation of full-length (Cep162-FL) or truncated (Cep162-N, Cep162-C) Cep162 proteins used for interaction assays in B. (B) Cep131 interacts with Cep162-FL and Cep162-C, but not Cep162-N in the Y2H assay. The upper panel shows the presence of Y2H plasmids as evidenced by colony growth on SD-Leu-Trp plates. The lower panel shows the positive interaction between Cep131 and Cep162 as evidenced by colony growth on SD-Ade-Leu-Trp-His plates. (C) Immunostaining of Cep162::GFP (green) in spermatocyte cilia in *cep131* mutants. Quantification of Cep162 signal intensity at the ciliary base is shown on the right. γ-Tubulin (red) labels the centriole/basal body. The error bars represent the mean ± SD, *n* = 30. (D) Immunostaining of Cep162::GFP (green), Actin

(red), and 21A6 (blue) in auditory cilia or olfactory cilia in WT or *cep131* mutants. Quantification of Cep162 signal intensity at the ciliary base is shown on the right. The error bars represent the mean ± SD, *n* = 30. (E) The subcellular localization of exogenous Cep162::GFP during spermatogenesis. From spermatogonia to late spermatocytes, Cep162 is localized at the tip of centriole/basal body. In round spermatids, as flagella elongates and the ciliary cap moves away from the BBs, Cep162 migrates with the ring centriole labeled by γ-Tubulin (arrowhead) and no signal is maintained at the BBs (arrow). During subsequent spermatid flagellar elongation, Cep162 signal disappears from the ciliary cap base labeled with Fbf1 (a transition fiber protein, red). γ-Tubulin (red) labels the centriole/basal body, axoneme is marked with Ac-Tub (magenta) and nuclei are marked with DAPI (blue). (F) The localization of GFP-tagged Cep162 N-terminus (1–447 aa) and C-terminus (448–897 aa) in spermatocytes. γ-Tubulin was used to label the BBs (red). (G) 3D-SIM images of Cep131:: GFP, Cep162::GFP, Cep290-C::GFP, or Cep290-N::GFP co-immunostained with antibody against Dzip1 (red). The plots of the signal intensity are shown on the right, respectively. (H) Graph showing the radial diameter of Cep131, Cep162, Cep290-N, and Cep290-C. (I) Schematic diagrams of localization pattern of Cep131 and Cep162 in the cross section of TZ. Scale bars: 5 μm (C, D), 2 μm (F), 500 nm (G), Zoom, 1 μm (D, E). The data underlying this figure can be found in S1 Data. BB, basal body; TZ, transition zone; WT, wild type.

Cep162 suggests a specific role in the initiation stage of ciliogenesis, but not as a constitutive component of the mature TZ. In addition, we noticed that unlike other TZ proteins, Cep162 was localized to the distal end of centrioles in spermatogonium (Fig 2E), suggesting that it was recruited to the centriole before cilia formation.

## Cep162 acts downstream of Cep131 to regulate ciliogenesis

To elucidate the role of Cep162 in fly ciliogenesis, we designed 2 gRNA to knockout Cep162 using the CRISPR-Cas9 system. We obtained a deletion mutant line, *cep162*[1] (c.981-1306Del), in which the C-terminus of Cep162 was lost due to reading frame shift caused by the deletion (Figs 3A, S4A and S4B). *cep162*[1] mutants were viable, but showed defects in cilia-related behaviors such as movement and hearing, which could be effectively rescued by expression of Cep162 (Fig 3B). Examination of the cilia morphology in auditory organ showed that about 20.2% of cilia were missing or very short in *cep162* mutants (Fig 3C). In addition, TEM analysis showed that missing spermatids were frequently observed in the cysts of *cep162* testes (Fig 3D). Therefore, Cep162 is indeed a key component for ciliogenesis in *Drosophila*. Notably, our TEM images revealed evident mitochondrial abnormalities in *cep162* spermatids, with some axonemes displaying very small or entirely lost mitochondrial derivatives (highlighted in Fig 3D). Additionally, the sizes of mitochondrial derivatives in the *cep162* mutant vary, contrasting with the overall even size observed in WT. How Cep162 affects mitochondria dynamics remain unclear. Interestingly, a recent study by Bauerly and colleagues reported the impact of cilia-related gene on mitochondria dynamics during *Drosophila* spermatogenesis [45]. Mutants in dynein-related ciliary genes exhibited the absence or reduced size of minor mitochondrial derivatives. Therefore, the influence of cilia-related genes on mitochondria is not a singular occurrence. While the underlying mechanism is presently unclear, further investigating is needed.

In spermatocyte cilia of *cep162* mutants, similar to *cep131* mutants, there was a significantly reduction in the signal intensities of Cep290, Dzip1, Cby, and TZ proteins Mks1 and Mks6 (Fig 3E). Additionally, we observed abnormally extended CG6652::GFP signals in 12.6% of spermatocyte centrioles (S4C Fig), indicating impaired BB docking in *cep162* mutants. Live imaging of the connection between the BB and the plasma membrane further confirmed the defective BB docking in some round spermatids (S4D Fig). Importantly, we found that the BB localization of Cep131 was normal in *cep162* mutants (Fig 3E). Collectively, our data indicate that *cep162* and *cep131* mutants exhibit similar phenotypes, with Cep131 being necessary for recruiting Cep162, whereas the reverse is not true.

## Cep162 is required for the correct localization of C-terminus of Cep290

We then asked whether Cep162 is the downstream protein of Cep131 responsible for regulating the localization of Cep290 C-terminus to the TZ. In fact, the signal of overexpressed

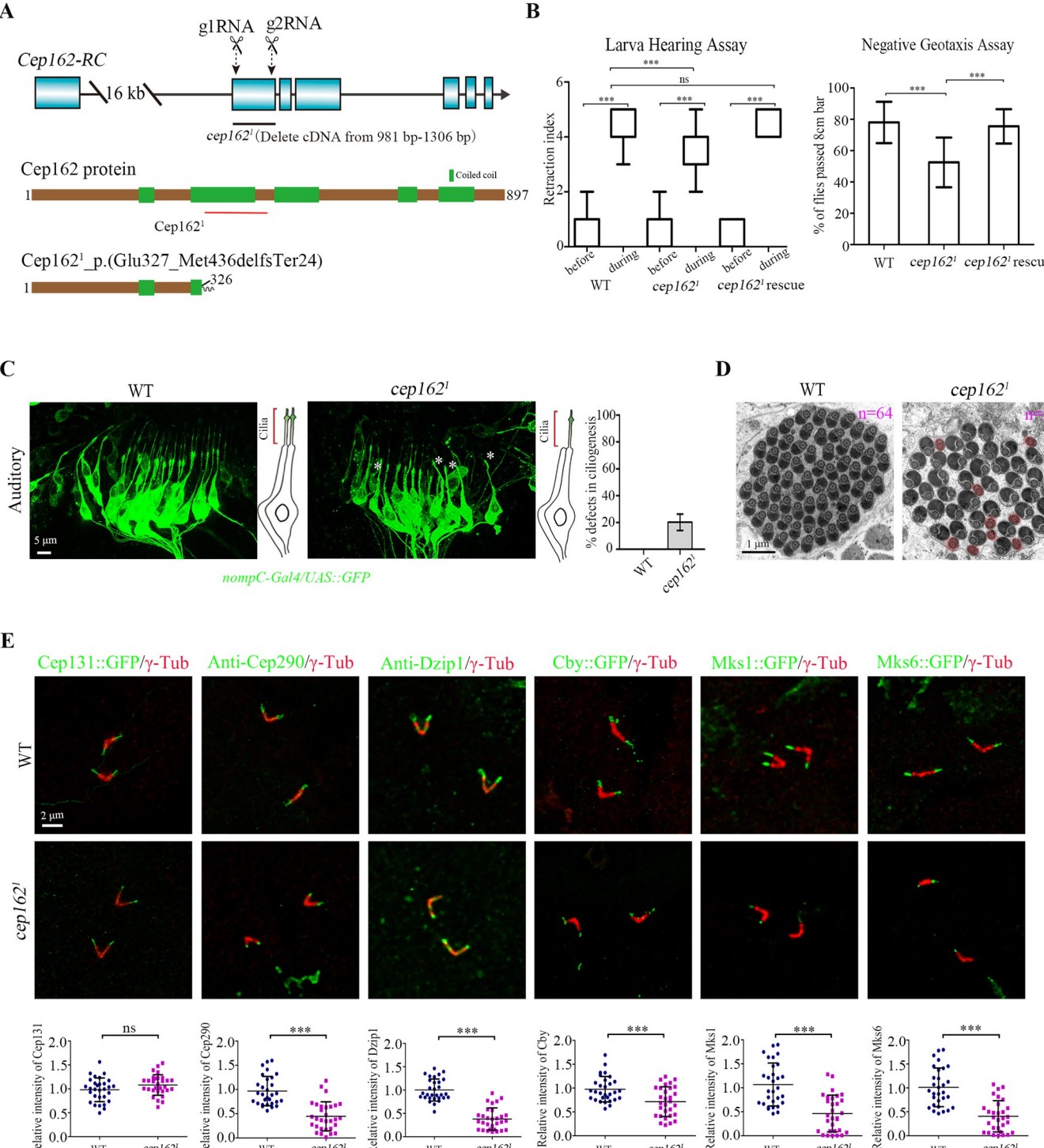

**Fig 3. *cep162* mutant mimics the phenotype of *cep131* mutant.** (A) Generation of *cep162* deletion mutants. Schematics show the genomic (upper panel) and protein (lower panel) structures of Cep162, along with the predicted protein products of *cep162¹* mutant (Cep1621_p.(Glu327_Met436delfsTer24)). Arrows point to 2 gRNA targeting sites. *cep162¹* mutant has a deletion in cDNA from nt 981 to 1306, resulting in a reading frame shift and C-terminus loss. (B) Analysis of hearing and negative geotaxis of *cep162* mutants. *cep162¹* flies show mild hearing defects. The retraction index indicates the larval response to a 1k Hz tone. The box shows the median and interquartile range; *n* = 25. The percentage of *cep162¹* flies passing the 8 cm high scale was significantly lower than that of WT flies. The error bars represent the mean ± SD, *n* = 50. (C) Living images of cilia morphology in antennal auditory organ of WT fly and *cep162* mutant pupae. Sensory neurons were labeled by *nompC*-Gal4/UAS::GFP (green), cilia are localized at the tip of dendrites. Sensory cilia are lost in partial sensory neurons (white asterisks) of *cep162¹* mutant. The graph on the right shows the percentage of sensory neurons with ciliary defects. (D) Representative TEM

images of elongating spermatid cysts in WT and *cep162* mutants. There are 64 spermatids per cyst in WT, whereas the number of spermatids per cyst is reduced in *cep162* mutant. (E) Immunostaining of Cep131, Cep290, Dzip1, Cby, Mks1, and Mks6 in WT or *cep162¹* testis. The quantification of the TZ protein intensities is shown on the lower panel. Unlike other TZ proteins, the localization of Cep131 in the TZ is normal. The error bars represent the mean ± SD, *n* = 30. Scale bars, 5 μm (C), 1 μm (D), 2 μm (E). The data underlying this figure can be found in S1 Data. TZ, transition zone; WT, wild type.

Cep290-C::GFP in *cep162* mutant spermatocytes was significantly reduced compared to WT (Fig 4A), but a certain amount of signal could still be observed. Notably, such residual Cep290-C::GFP signal was much stronger than that observed in *cep131* mutants (Figs 1A and 3E). Mammalian Cep290 C-terminal fragment was previously shown to interact with itself or to Cep290 N-terminal fragment, forming homodimers or heterodimers [46]. As part of endogenous Cep290 was still able to localize to the TZ in *cep162* mutants, we therefore speculated that overexpressed Cep290-C::GFP might bind to remaining endogenous Cep290. To exclude this possibility, we generated the *cep162* and *cep290¹* double mutant. Previously, we have shown that the TZ assembly is completely blocked and that centriole/basal body microtubules extend abnormally in *cep290¹* mutant [29]. Interestingly, we observed that both Cep162-FL:: GFP and Cep162-C::GFP were localized along the abnormally extended microtubules in *cep290¹* single mutant (Fig 4B), suggesting that Cep162 can recognize microtubules independently of Cep290. Intriguingly, in *cep290¹* single mutants, Cep290-C::GFP showed a similar localization pattern as Cep162::GFP along the abnormally extended microtubules (Fig 4A), demonstrating that Cep290-C::GFP does not need full length Cep290 to be targeted to the axoneme. Furthermore, the signal of Cep290-C::GFP was significantly reduced in spermatocytes of *cep162; cep290¹* double mutants (Fig 4A), despite the persistence of abnormal microtubule extensions indicated by Ac-tub (S4E Fig) in these double mutants. This observation suggests that Cep162 regulates the association between Cep290 C-terminus with microtubules in spermatocyte cilia. Collectively, our results indicate that the localization of Cep290-C to the axoneme is mediated by Cep162, while it can still be retained by endogenous Cep290 in *cep162* mutants. Similar results were also observed in sensory cilia, where the deletion of Cep290 did not affect the targeting of Cep162-FL or Cep162-C to the ciliary base in sensory neurons (Fig 4C), but Cep290-C::GFP was missing from the TZ in *cep162; cep290¹* sensory neurons (Fig 4D).

## Cep162 genetically interacts with Cby-Fam92 module to initiate ciliogenesis

Since ciliogenesis initiation was completely abolished in *cep131* and *cby* double mutants [43], we speculated that *cep162* and *cby* double mutants should have a similar phenotype. Indeed, *cep162; cby* flies showed much more severe cilia-related defects than either single mutant. The *cep162; cby* flies were severely uncoordinated, unable to walk and fly. Hearing assay indicated that hearing was completely lost in double mutants (Fig 5A). Morphological examination of auditory cilia revealed a failure to form cilia in auditory organ (Fig 5B). In spermatocytes, the percentage of abnormal extensions of the centriole microtubules, labeled by CG6652, increased from 12.6% in *cep162* single mutants, or 47.6% in *cby* single mutants to 81.1% in *cep162; cby* double mutants, indicating a strong synthetic defect in the initiation of ciliogenesis (Fig 5C). In addition, similar to *cep131; cby* double mutants, sperm flagella in *cep162; cby* spermatids were severely affected, with almost no axoneme observed in TEM analysis (Fig 5D). Consistent with this observation, the signals of Cep290, Dzip1, Mks1, and Mks6 were completely lost from the tips of BBs in the double mutants (Fig 5E and 5F). All these results indicate that the *cep162*; *cby* mutants mimic the phenotype of *cep131; cby* double mutants previously reported.

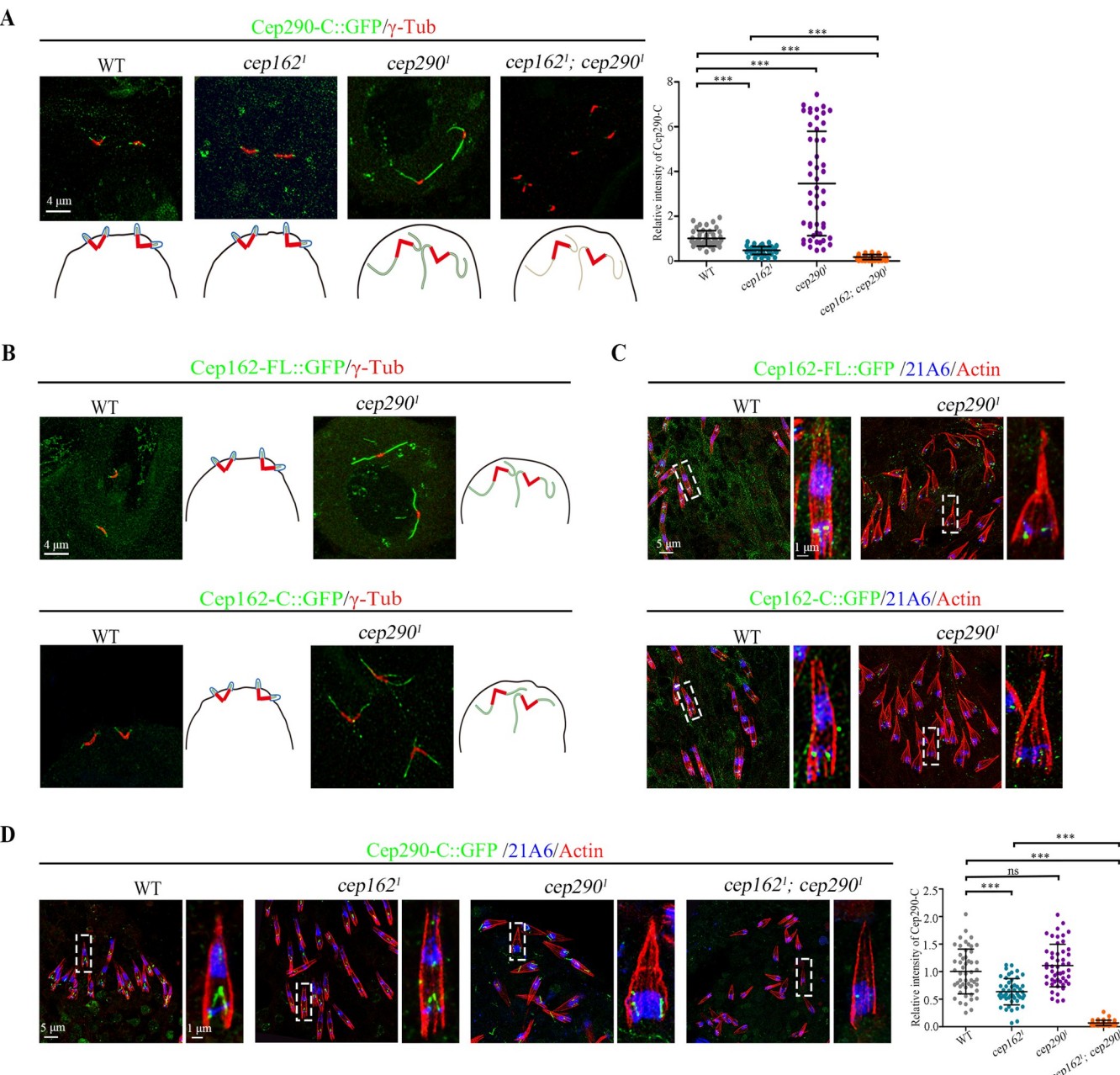

**Fig 4. Cep162 is required for the BB localization of Cep290 C-terminus.** (A) Immunostaining of Cep290-C::GFP (green) in spermatocyte cilia of WT, *cep162*, *cep290¹*, and *cep162; cep290¹* flies, and the quantification of their corresponding relative fluorescence intensities is shown on the right. Cep290-C::GFP completely lose TZ localization in the *cep162* and *cep290¹* double mutant spermatocytes. Centriole/basal body is marked with γ-Tubulin (red). The error bars represent the mean ± SD, *n* = 50. (B) Immunostaining of Cep162-FL::GFP (green) and Cep162-C::GFP (green) in WT or *cep290¹* spermatocyte cilia. The cartoon shows Cep162 signals in WT or *Cep290¹*. Centriole/basal body is marked with γ-Tubulin (red). (C) Cep162 signals are grossly normal in *cep290¹* mutant antennae. 21A6 (blue) marks the cilia base; Actin (red) marks the ciliated region. (D) Immunostaining of Cep290-C::GFP (green) in WT, *cep162*, *cep290¹*, and *cep162; cep290¹* antennae, and the quantification of their corresponding relative fluorescence intensities is shown on the right. Notably, Cep290-C::GFP signal is significantly reduced in *cep162; cep290¹* double mutants. 21A6 (blue) marks the ciliary base; Actin (red) marks the cilia region. The error bars represent the mean ± SD, *n* = 50. Scale bars, 4 μm (A, B), 5 μm (C, D), Zoom, 1 μm (C, D). The data underlying this figure can be found in S1 Data. BB, basal body; WT, wild type.

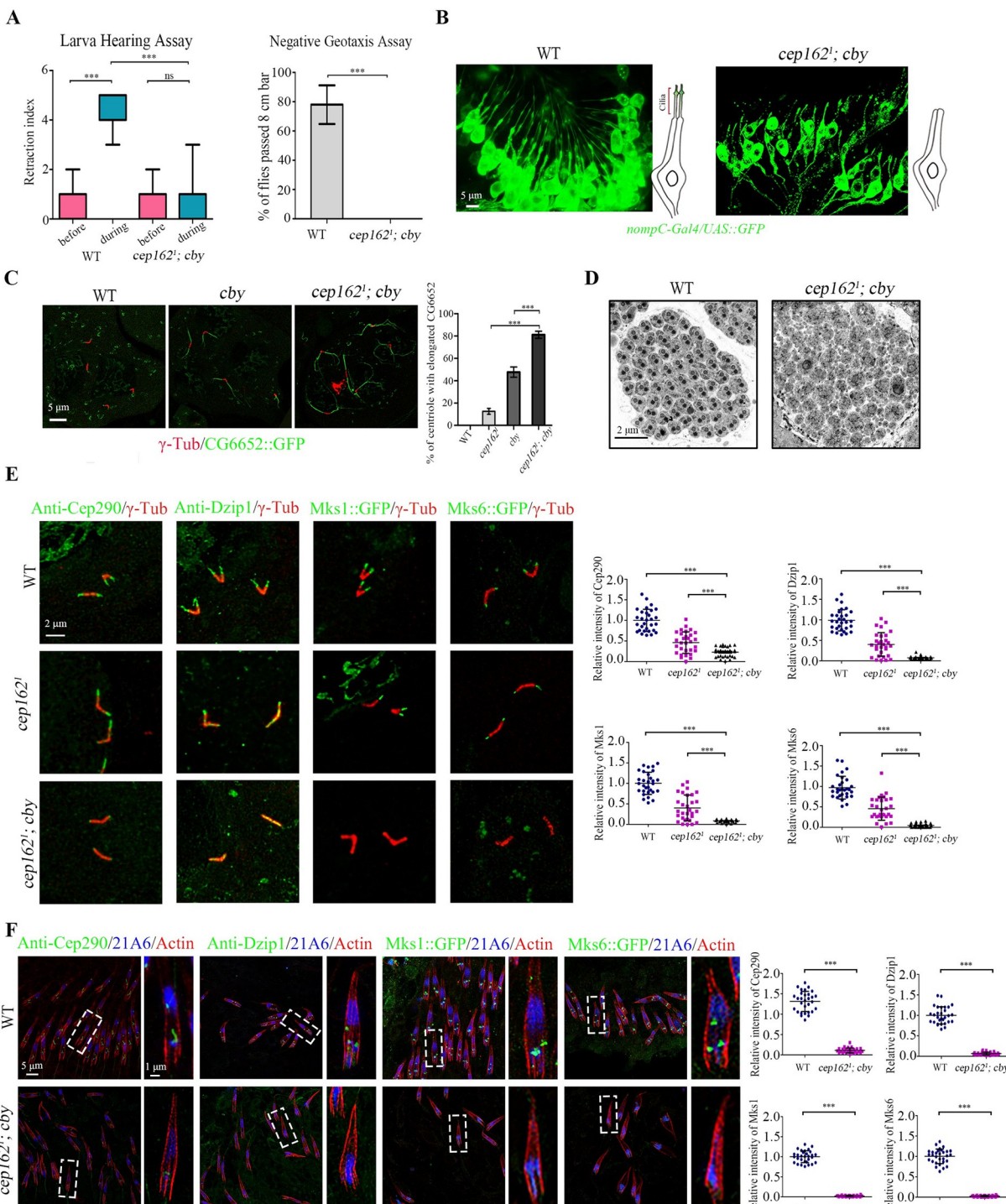

**Fig 5. Cep162 genetically interacts with Cby-Fam92 module to initiate ciliogenesis.** (A) *cep162*; *cby* flies completely lose their hearing and negative geotaxis. (B) Living images of cilia morphology in pupal antennal auditory neurons in WT flies and *cep162; cby* mutants. Cilia are completely lost in *cep162; cby* mutants. Sensory neurons are labeled by *nompC*-Gal4/UAS::GFP (green), and cilia are localized at the tip of dendrites. (C) Compared to WT or *cby* single mutants, in *cep162; cby* mutants, the percentage of spermatocyte cilia with aberrant extensions of CG6652 signal is significantly increased. Basal bodies are marked with γ-Tubulin (red). (D) Compared to WT which has 64 flagella in each spermatid cyst, few normal flagella are observed in spermatid cysts of *cep162; cby* mutants. (E) Cep290, Dzip1, Cby, and Mks1 are absent from the TZ in *cep162; cby* mutant spermatocytes. Right panel show the quantification of corresponding relative fluorescence intensities. Centriole/basal body is marked with γ-Tubulin (red). The error bars represent the mean ± SD, *n* = 30. (F) Cep290, Dzip1, Cby, and Mks1 are absent from the TZ in *cep162; cby* mutant sensory neuron. Right panel show the quantification of corresponding relative fluorescence intensities. 21A6

(blue) marks the ciliary base; Actin (red) marks the ciliated region. The error bars represent the mean ± SD, $n$ = 30. Scale bars, 5 μm (B, C, F), 2 μm (D, E), Zoom, 1 μm (F). The data underlying this figure can be found in S1 Data. TZ, transition zone; WT, wild type.

As Cby and Fam92 function together in a module to regulate ciliogenesis [30,47,48], we speculated that the combined mutation of Cep162 with Fam92 might also lead to synthetic ciliary defects. Indeed, as expected, the *cep162; fam92* flies showed much more severe defects in walk and fly than either single mutant alone, and Cep290 was also completely lost in spermatocyte cilia of the double mutants (S5 Fig).

## Cby-Fam92 module is required for the association of the N-terminus of Cep290 with the membrane

The synthetic defects observed in Cep162/Cep131 and Cby/Fam92 double mutants could likely be attributed to the complete loss of Cep290 signal at the BB. Considering that Cep131-Cep162 module is required for the localization of Cep290 C-terminus, and the localization pattern of Cep290 N-terminus is similar to that of Cby near the membrane, we speculated that Cby-Fam92 module might play a role in promoting the BB localization of Cep290 N-terminus. To exclude the effect of Cep290 C-terminus on our analysis, we combined deletions of Cep290 C-terminus (*cep290$^{\Delta C}$*) with Cby, and checked the localization of overexpressed Cep290-N::GFP. Indeed, the TZ signal of Cep290-N::GFP was almost completely lost in *cby; cep290$^{\Delta C}$* double mutants compared with that in *cby* or *cep290$^{\Delta C}$* single mutants (Fig 6A), indicating that Cby does play a role in targeting the Cep290 N-terminus to the TZ. Notably, unlike Cep290-N:: GFP, Cep290-C::GFP was still able to localize to the tips of basal bodies in *cby; cep290$^{\Delta C}$* double mutants (Fig 6B), suggesting that Cby has a specific role on the localization of Cep290-N::GFP.

Given that Cby affects the localization of Cep290 N-terminus, we reasoned that endogenous truncated Cep290 may not be able to target to TZ in *cby; cep290$^{\Delta C}$* double mutants. Therefore, we examined the Cep290 signal in spermatocytes using our Cep290 antibody. In *cby* single mutant, Cep290 signal was slightly decreased. In *cep290$^{\Delta C}$* single mutant, Cep290 N-terminus is expressed at lower levels as previously described [29] but is still retained at the tip of BB. However, consistent with our hypothesis, this TZ signal of remaining Cep290 truncated form was completely lost in spermatocytes of *cby; cep290$^{\Delta C}$* double mutants (Fig 6C). Similar results were also observed in sensory cilia, where Cep290 signal was also completely lost in *cby; cep290$^{\Delta C}$* double mutants (Fig 6D). As well, we demonstrated that Cep290 signal was also completely lost in *fam92; cep290$^{\Delta C}$* double mutants (S5 Fig). All these results indicated that Cby-Fam92 module specifically promotes the anchoring Cep290 N-terminus to the membrane.

Since Cep290 was completely lost from BBs in *cby; cep290$^{\Delta C}$*, the initiation of ciliogenesis should be completely blocked like in *cby; cep131* or *cby; cep162* double mutants. In fact, as expected, the proportion of abnormal extensions of CG6652 at the tips of the centrioles reached about 79.8%, indicating that most of the BBs do not anchor to the plasma membrane (Fig 6E). In addition, Dzip1 and Mks1 were completely lost from the tips of BBs in both ciliated cell types (Fig 6F and 6G). All these results indicate that the *cby; cep290$^{\Delta C}$* mutant mimics the phenotype of *cep290* null mutant.

## Discussion

The TZ protein Cep290 bridges the ciliary membrane and the axonemal microtubules with its N-terminus close to the membrane and its C-terminus close to the axonemal microtubules [33,34]. Previous studies by us and others have shown that the N-terminus of Cep290 acts

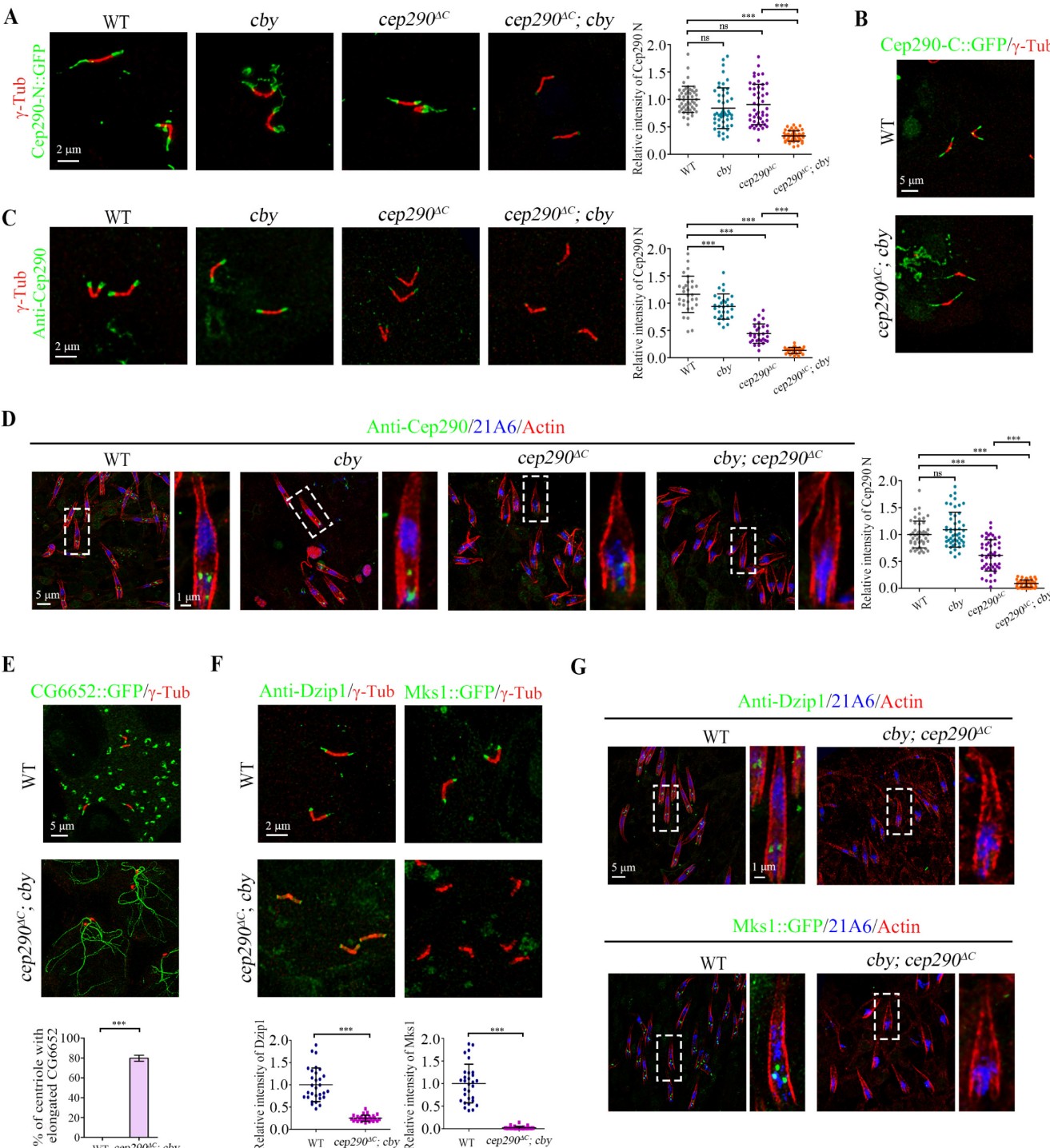

**Fig 6. The Cby-Fam92 module is required for the BB localization of the N-terminus of Cep290.** (A) Immunostaining of the exogenous Cep290-N::GFP (green) in spermatocyte cilia of WT, *cby*, *cep290*$^{ΔC}$, *cby; cep290*$^{ΔC}$ testis and the quantification of corresponding relative fluorescence intensities is shown on the right. The signal of Cep290-C::GFP is almost lost at the tips of centrioles in *cby; cep290*$^{ΔC}$ double mutants. Centriole/basal body is marked with γ-Tubulin (red). The error bars represent the mean ± SD, *n* = 60. (B) Immunostaining of the exogenous Cep290-C::GFP (green) in spermatocyte cilia of WT and *cby; cep290*$^{ΔC}$ testis. Cep290-C::GFP was localized to the axoneme in *cby; cep290*$^{ΔC}$ mutants. Centriole/basal body is marked with γ-Tubulin (red). (C) Immunostaining of the endogenous Cep290 in WT, *cby*, *cep290*$^{ΔC}$, and *cby; cep290*$^{ΔC}$ testis and the quantification of corresponding relative fluorescence intensities were shown on the right. The C-terminal truncated Cep290 form fails to locate to the tips of centrioles in *cby; cep290*$^{ΔC}$ spermatocytes. Centriole/basal body is marked with γ-Tubulin (red). The error bars represent the mean ± SD, *n* = 30. (D) The endogenous Cep290 C-terminal truncated form fails to locate to the cilia base in

antennal auditory neurons of *cby; cep290^{ΔC}* antenna. Right panel shows the quantification of corresponding relative fluorescence intensities. 21A6 (blue) marks the cilia base; Actin (red) marks the ciliated region. The error bars represent the mean ± SD, *n* = 60. (E) In *cby; cep290^{ΔC}* mutants, the percentage of spermatocyte cilia with aberrant extension of CG6652 signal reaches about 79.8%. Axoneme is marked with CG6652. Centriole/basal body is marked with γ-Tubulin (red). The error bars represent the mean ± SD, *n* = 30. (F) Dzip1 and Mks1 are absent from the TZ in spermatocyte cilia of *cby; cep290^{ΔC}* mutants. Centriole/basal body is marked with γ-Tubulin (red). The error bars represent the mean ± SD, *n* = 30. (G) Dzip1 and Mks1 fail to locate to the cilia base in antennal auditory neurons of *cby; cep290^{ΔC}* antenna. 21A6 (blue) marks the cilia base; Actin (red) marks the ciliated region. Scale bars, 2 μm (A, C, F), 5 μm (B, D, E, G), Zoom, 1 μm (D, G). The data underlying this figure can be found in S1 Data. BB, basal body; TZ, transition zone; WT, wild type.

upstream of Dzip1-Cby-Fam92 module and is essential for ciliogenesis initiation and ciliary bud formation in *Drosophila* [29]. Here, we discover that Cep131-Cep162 module functions upstream of Cep290 and regulates the association of Cep290-C-terminus with axonemal microtubules to initiate ciliogenesis. Taken together, we propose that the initial process of ciliogenesis in *Drosophila* spermatocyte is as follows (Fig 7A and 7B): in spermatogonia, Cep131 localizes to the distal end of centriole and recruits Cep162 to the centriole; when the centriole starts to grow a cilium in spermatocytes, Cep162 recruits Cep290 and promotes the binding of Cep290-C to the axoneme; subsequently, the conformation of Cep290 changes from a closed to an open state, and the N-terminus of Cep290 recruits Dzip1-Cby-Fam92 module to start early ciliary membrane formation and ciliary bud formation. On the other hand, Cby-Fam92-mediated ciliary membrane formation has a positive feedback effect on promoting the association of Cep290 N-terminus with the ciliary membrane. Given that similar results were observed in sensory cilia, this ordered model is not exclusive to spermatocyte cilia. Moreover, all proteins (Cep131, Cep162, Cep290, Cby, Fam92) involved in this ordered pathway are conserved and play a critical role in TZ assembly in mammals [24,37,44,48], suggesting a potential conservation of this model in mammals. However, it should been noted that Cep290 has been shown to have different properties across different species [9], and certain animal models carrying mutations associated with patients have displayed milder symptoms [49,50], tissue and species-specific functions of Cep290 can not be ruled out.

Previously, Drivas and colleagues have proposed a conformation change model of Cep290 during ciliogenesis [33]. According to their model, Cep290 is initially maintained in a closed and inhibited state by its N and C termini [33]. During ciliogenesis, Cep290 undergoes a conformational change, transitioning from a closed to an open state. This conformational change allows Cep290's membrane-binding and microtubule-binding domains to become accessible, enabling the recruitment of additional interacting partners to initiate ciliogenesis. The validity of this conformational change model is supported by 2 pieces of evidence. Firstly, the N-terminal fragment of Cep290 was found to interact with the C-terminal fragment of Cep290 in vertebrate [46], providing evidence for their association. Secondly, both the N and C termini of Cep290 have been shown to play regulatory roles in the localization and function of Cep290 in both mammals and *Drosophila* [29,33]. Interestingly, we observed that the diameter of Cep290-N::GFP in *cep131* mutants were significantly smaller than that in WT (S1E Fig), suggesting that the conformation of Cep290 may still be closed in *cep131* mutants, implying the involvement of Cep131 in conformation change of Cep290. However, direct structural evidence for the conformational change of Cep290 is currently lacking. Investigating this aspect and exploring the role of Cep131 in it will be a promising research direction in the future.

Notably, Cep290 has its own microtubule-binding domain and membrane-binding amphipathic α-helix motif [33]; therefore, the role of Cep131-Cep162 and Cby-Fam92 modules might just be to facilitate and enhance the efficiency of Cep290 connection with the axoneme and the membrane. Given the partial defects in ciliogenesis in the absence of Cep131 or Cep162, we speculate that the microtubule binding ability of Cep290 may be sufficient to build the TZ in absence of either Cep131 or Cep162. We propose that as long as Cep290, even in

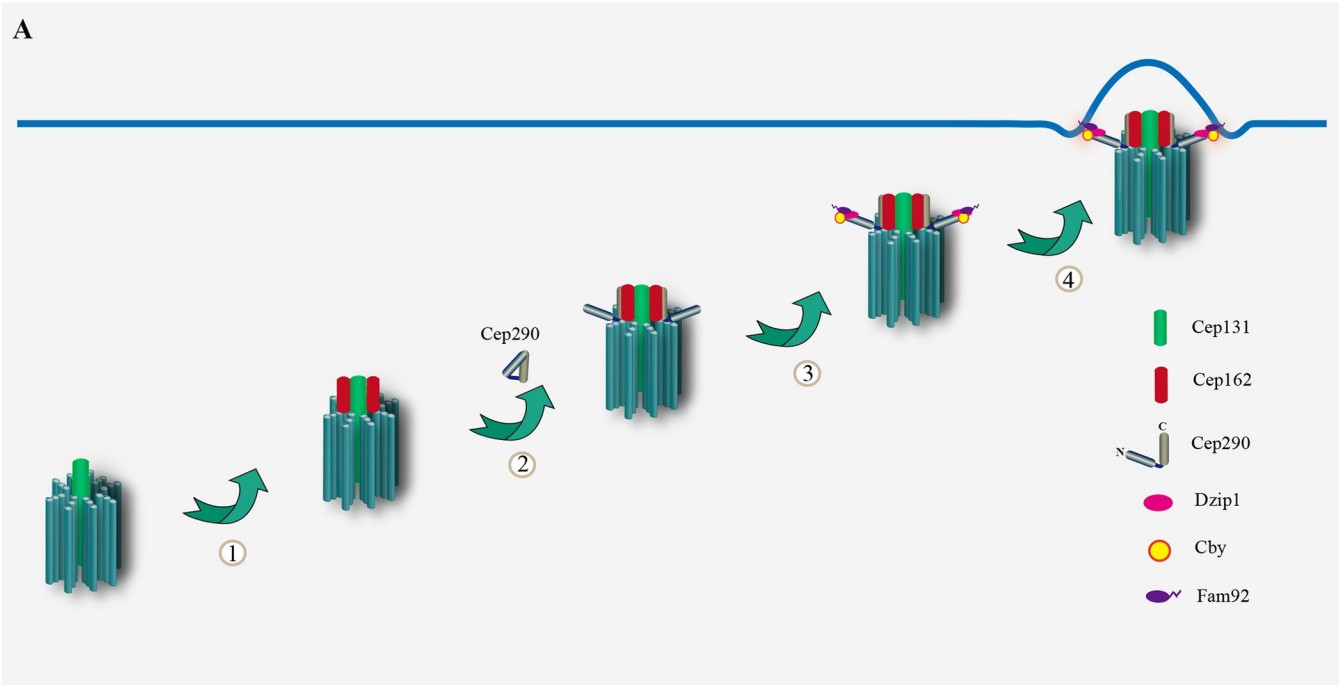

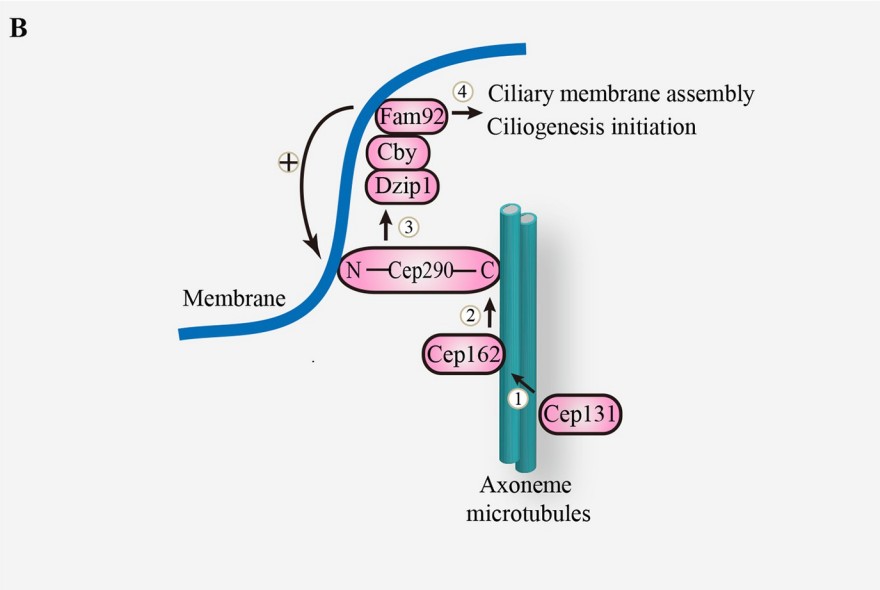

**Fig 7. Models for ciliogenesis initiation and the mechanism of CEP290 localization in *Drosophila*.** (A) Model for ciliogenesis initiation in *Drosophila*. ① In spermatogonia, Cep131 localizes to the distal end of centriole and recruits Cep162. ② When the centriole starts to grow cilia in spermatocytes, Cep162 recruits Cep290 and promotes the binding of Cep290-C to the axoneme. ③ Subsequently, the conformation of Cep290 changes from a closed to an open state, and the N-terminus of Cep290 recruits Dzip1-Cby-Fam92 module; ④ and then start early ciliary membrane formation and ciliary bud formation. (B) The cooperative model for Cep290 localization. Cep131-Cep162 module together with Cby-Fam92 module regulates the localization of Cep290 at the TZ. Mechanistically, Cep131 recruits Cep162 which mediates the association of C-terminus of Cep290 to microtubule. The N-terminus of Cep290 recruits Dzip1-Cby-Fam92 module to start early ciliary membrane formation and ciliary bud formation, whereas Cby-Fam92-mediated ciliary membrane formation has a positive feedback effect on promoting the association of Cep290 N-terminus with the ciliary membrane. TZ, transition zone.

limited amounts, is initially able to localize to the BB, initiation of ciliogenesis can then proceed through the mutual recruitment of Cep290 and Dzip1-Cby-Fam92 module. This is supported by our previous observation that even the N-terminus of Cep290 alone, when properly localized, can promote cilliogenesis initiation [29]. This model provides a possible explanation for the wide range of pathological phenotypes associated with mutations in Cep290, ranging from isolated blindness to lethality, making it challenging to establish a clear genotype–phenotype correlation. We anticipate that particular developmental and cellular context, or the presence of genetic modifiers may introduce variability in cilium assembly or function by affecting various complexes required to stabilize Cep290 at the TZ.

Our work not only uncovers the molecular mechanism of the synthetic interaction between Cby-Fam92 module and Cep131-Cep162 module in Cep290 recruitment and ciliogenesis initiation, but also reveals a novel molecular function of Cep131 in ciliogenesis, which recruits Cep162 to promote the binding of the C-terminus of Cep290 to the axoneme. We demonstrated that the Cep131-Cep162 module promotes Cep290 C-terminus binding to the axoneme, whereas Cby-Fam92 module promotes and stabilizes the localization of Cep290 N-terminus on the membrane, therefore, the concurrent mutation of these 2 modules collectively leads to the failure of Cep290 to localize to the TZ and blocks the initiation of ciliogenesis. Cep290 is an intriguing ciliopathy gene. More than 130 mutations have been identified, but their associated phenotypes can be dramatically different, suggesting that there may be genetic modifiers involved in the development of their phenotypes [32,51]. Therefore, identification of the modifier genes has become an important element to understand the pathogenesis of Cep290-related ciliopathies. While Cep131 and Cby have not yet been associated with ciliopathy, mutations in Cep162 and Fam92A have been reported to be linked to human cilia-related diseases [52,53]. Our observation of the synthetic defects in ciliogenesis initiation in *cby; cep290^{ΔC}* double mutants suggests that Cby or Fam92 may be genetic modifiers of disease mutations in Cep290 C-terminus. Similarly, it is possible that Cep131 or Cep162 may serve as genetic modifier for disease mutations in Cep290 N-terminus.

## Materials and methods

### Fly stocks

Transgenic flies of Cby::GFP, Cep131::GFP, Cep290-N::GFP, Cep290-C::GFP, CG6652::GFP, Mks1::GFP, Mks6::GFP, and *nompC-Gal4;UAS::GFP* were previously reported [29]. Cep162::GFP, Cep162-N::GFP and Cep162-C::GFP transgenic flies were generated in this paper. The cDNA of Cep162 or Cep162 C-terminus and its endogenous promoter were cloned and inserted into the PJFRC2 vector, and the plasmids were then used to construct transgenic flies by the Core Facility of *Drosophila* Resources and Technology, Shanghai Institute of Biochemistry and Cell Biology (SIBCB), Chinese Academy of Sciences (CAS).

$w^{1118}$ flies were used as wild type. *cep290^{ΔC}*, *cep290^1*, *cby* and *fam92* mutant flies have been described previously [29]. All experiments were conducted at 25˚C.

### Generation of deletion mutants in *Drosophila*

Generation of *cep131* and *cep162* mutants were performed as previously reported [29]. Briefly, the mutant for Cep131 and Cep162 were generated by the CRISPR/Cas9-mediated gene targeting system. The gRNA expression plasmids were generated by inserting the targeting sequences into the PU6-BbsI-chiRNA vector using PCR. To increase the efficiency of generating fragment deletion mutants that facilitate mutant identification, 2 gRNAs were injected together into Cas9-expressing embryos.

Primers used to construct gRNA vector:

Cep162 g1 F: 5′-GTCGCTCTGAAGGCGAACGGGTTTTAGAGCTAGAAATAGC-3′;
Cep162 g1 R: 5′-CCGTTCGCCTTCAGAGCGACCGACGTTAAATTGAAAATAGG-3′;
Cep162 g2 F: 5′-TCAGTATGCGCTCCATCTCGGTTTTAGAGCTAGAAATAGC-3′; Cep162
g2 R: 5′-CGAGATGGAGCGCATACTGACGACGTTAAATTGAAAATAGG-3′; Cep131 g1
F: 5′-CAAGCACAAGCCAGGACTGG GTTTTAGAGCTAGAAATAGC-3′: Cep131 g1 R: 5′-
CCAGTCCTGGCTTGTGCTTGCGACGTTAAATTGAAAATAGG-3′: Cep131 g2 F: 5′-
CTCCCTCTGCGAGAAGGTGGGTTTTAGAGCTAGAAATAGC-3′: Cep131 g2 R: 5′-
CCACCTTCTCGCAGAGGGAGCGACGTTAAATTGAAAATAGG-3′.

Primers used to identify mutants:
Cep162 F: 5′-ATATTTTCGCGAGCTGAGGACAC-3′;
Cep162 R: 5′-GCGATGTGAGTCTCATATTTGGC-3′;
Cep131 F: 5′-CATCAGTGTGGGCAGCCTACG-3′;
Cep131 R: 5′-GCGAATGCTAGTCTCGATCTGC-3′.

## Immunofluorescence

For IF staining of antennae or testes, 36 to 48 h after puparium formation (APF), pupae were collected and their antennae or testes were dissected with forceps in PBS. Antennae or testes were transferred to the center of a coverslip, and then gently covered by a slide over the coverslip. The slide was dipped into liquid nitrogen for 30 s, and the coverslip was immediately removed with a blade. The specimens were then fixed using methanol (−20°C) for 15 min, followed by acetone (−20°C) for 10 min. To block nonspecific binding, the specimens were incubated for 1 h in blocking buffer (0.1% Triton X-100, 3% bovine serum albumen in PBS). The primary antibodies were applied overnight in a moisture chamber at 4°C, and then the secondary antibodies were applied for 3 h at room temperature.

## Antibodies

The primary antibodies used were as follows: Rabbit anti-Dzip1 (aa 451–737), Rabbit anti-Cep290 (aa 292–541), and Rabbit anti-Fbf1 (aa 1–336) antibodies were generated at YOUKE Biotech, rabbit anti-GFP (1:500, ab290, Abcam), mouse anti-GFP (1:200, 11814460001, Roche), mouse anti-Ac-tub (1:500, T6973, Sigma-Aldrich), mouse anti-γ-Tubulin (1:500, T5326, Sigma-Aldrich), and mouse anti-21A6 (1:200, AB528449, DSHB). The following secondary antibodies were used: goat anti-mouse Alexa Fluor 488 (1:1,000, A-11001, Invitrogen), goat anti-rabbit Alexa Fluor 594 (1:1,000, A1000701, Invitrogen), goat anti-rabbit Alexa Fluor 488 (1:1,000, A-11006, Invitrogen), goat anti-mouse IgG1 Alexa Fluor 488 (1:1,000, A-21121, Invitrogen), goat anti-mouse Alexa Fluor 594 (1:1,000, A-11007, Invitrogen), goat anti-mouse Alexa Fluor 647 (1:1,000, A-21242, Invitrogen).

## Microscopy and image analysis

For IF staining, images were taken on a fluorescence microscope (Nikon Ti) with a 100× (1.4 NA) oil-immersion objective, or a confocal microscope (Leica Stellaris 5) with a 63× (1.4 NA) oil-immersion objective, or the Delta Vision OMX SR (GE Healthcare) with a 60× (1.42 NA) oil-immersion objective. Confocal images were acquired as Z-stacks (0.5 to 0.8 μm for Z-step size and 3 to 5 for number of steps) using xzy scan pattern. The sections of 3D-SIM images were acquired at 0.125 μm Z-steps (20 steps) and the raw data were reconstructed by using softWoRx software (GE Healthcare). Images were quantified for the pixel density using ImageJ (National Institutes of Health). For quantification of the pixel density, images were taken using equal microscopy settings. The pixel density values were calculated by the sum pixel density

values in a defined region subtracting the sum pixel density values in an area close to the defined region. All images assembled into figures using Photoshop (CS5, Adobe).

## Transmission electron microscopy

Samples were prepared for electron microscopy as previously described [54]. Briefly, samples were incubated in 2.5% glutaraldehyde/0.2 M phosphate buffer on ice for 24 h, postfixed in 2% $OsO_4$/0.1 M phosphate buffer on ice, dehydrated with ethanol, and embedded in epoxy resin. Selected areas were sectioned using an ultramicrotome. Ultrathin sections were stained with uranyl acetate and lead citrate, and examined with Hitachi H-7650 transmission electron microscopy at 80 kV.

## Negative geotaxis assay

Virgin flies were collected and cultured in fresh medium for 3 to 5 days. A total of 50 flies were sorted into 5 measuring vials of 10 each, and then tapping flies to the bottom of the vials and counting the number of flies that climbed over the 8 cm high bar within 10 s. Each group was repeated 3 times.

## Larval hearing assay

Third instar larvae were divided into 5 groups of 5, placed on an agar plate above the speaker, and stimulated with 1k Hz sound every 30 s, the number of larvae with contractile responds on the head or body within 1 s after stimulation were counted. Each group was repeated 5 times.

## Yeast two-hybrid assay

Cep131-FL (1–1114 aa), Cep131-N1 (1–480 aa), Cep131-M1 (481–781 aa), Cep131-C1 (782–1114 aa), Cep162-N (1–447 aa), Cep162-C (448–897 aa), Cep290-N (1–887 aa), and Cep290-ΔN (888–1978 aa) were introduced into either pGBKT7 or pGADT7 vector. Clones in pGADT7 and pGBKT7 were transformed into yeast strain AH109 (Takara Bio). The yeasts were grown on SD-Leu-Trp plates at 30˚C. After 3 to 4 days incubation, 2 independent positive colonies were picked and diluted with TE buffer (10 mM Tris-HCl, 1 mM EDTA (pH 7.5)), and then transferred to SD-Ade-Leu-Trp-His or SD-Leu-Trp-His plates with 3-amino-1,2,4-triazole and incubated for 5 days at 30˚C.

## GST pull-down assay

To generate bacterial expression plasmids for His-Cep162-N (1–447 aa), His-Cep162-C (448–897 aa), His-Cep131-N (1–549 aa), His-Cep131-C (550–1114 aa), the cDNA fragments encoding the indicated amino acids were amplified by PCR and subcloned into the pET28a or pGEX-4 T-1 vector. The proteins were expressed using BL21 (DE3) *E. coli*. strain with IPTG induction and purified with Glutathione-agarose beads or Ni-resin (Yeasen). Purified His-Cep131 truncations were incubated with immobilized GST or GST-Cep162 truncations in the binding buffer (25 mM Tris-HCl at pH 7.4, 150 mM NaCl, 0.5% Triton X-100, 1 mM dithiothreitol, 10% glycerol, and protease inhibitors) at 4˚C for 4 h. After incubation, the beads were washed 3 times with washing buffer (binding buffer with 20 mM imidazole) and then boiled for 10 min in 1 × SDS loading buffer. The protein samples were then separated by SDS-PAGE gels and transferred to the PVDF membrane for either immunoblotting with His antibody (Invitrogen) or staining with Ponceau S. Primary antibodies were used at a dilution of 1/1,000,

and secondary antibodies were used at a dilution of 1/2,000. All uncropped images can be found in S1 Raw Image.

## Statistics

Data were analyzed and graphed using Microsoft Excel or Graphpad Prism. Unless otherwise indicated, all error bars represent the standard deviation (SD) of the mean, and the statistical significance between data was assessed with an unpaired two-tailed Student's $T$ tests. Differences between data were considered statistically significant when $P \leq 0.05$.

## Supporting information

**S1 Fig. Identification and phenotype analysis of *cep131* mutants.** (A) Diagram showing of the generation of *cep131* mutants. Schematics show the genomic (upper panel) and protein (lower panel) structures of Cep131, along with the predict protein product of *Cep131[1]* mutant (Cep1311_p.(His480_Leu708delfsTer61)). Two arrows represent gRNA target sites. *cep131[1]*, a frameshift line, has a deletion in cDNA from nt 1439 to 2123, resulting in a reading frame shift and C-terminus loss. (B) Genotyping of *cep131* mutants using PCR. The PCR amplification products were 1471 bp long for *w[1118]* and 603 bp long for *cep131* flies. (C) Sequence confirmation of the deletion in *cep131* mutant. Primers used for sequence are marked with orange frames. The locations of 2 gRNAs used for mutant generation are underlined with black. Red and blue frames label the boundary of deletion *cep131* in mutant. (D) Immunostaining of CG6652 in spermatocyte cilia of WT or *cep131[1]* testis. CG6652 (green) marks the ciliary axoneme. In *cep131[1]*, a few centrioles have over elongated CG6652 signals (arrows). Centriole/basal body is marked with γ-Tubulin (red). Scale bars, 5 μm. (E) Quantification of the radial distance of Cep290 or Cep290-N. In *cep131* mutants, the radial distance of Cep290 or Cep290-N were significantly reduced compared to WT. The error bars represent the mean ± SD, $n = 60$. The data underlying this figure can be found in S1 Data.
(TIF)

**S2 Fig. Yeast two-hybrid assay of the interaction between Cep290 and Cep162 or Cep131.** (A) In Y2H assay, Cep290 interacts with Cep162 (CG42699), but not Cep131 in *Drosophila*. LW: Selective media SD-Leu-Trp plates. LWH: Selective media SD-Leu-Trp-His plates. LWHA: SD-Ade-Leu-Trp-His plates. (B) The GST pull-down assay confirmed the direct interaction between Cep131 and Cep162 in vitro. GST, GST-Cep162-N, and GST-Cep162-C recombinant proteins were pulled down with His-Cep131-N or His-Cep131-C proteins. (C) In Y2H assay, Cep162 interacts with Cep131-N1 and Cep131-C1, but not Cep131-M1. LW: Selective media SD-Leu-Trp plates. LWH: Selective media SD-Leu-Trp-His plates. LWHA: SD-Ade-Leu-Trp-His plates.
(TIF)

**S3 Fig. Protein sequence alignment of the C-terminus of human Cep162, mice Cep162, and *Drosophila* CG42699.** CG42699 is the only one significant alignment sequence when searching for homology of human or mice Cep162 protein in *Drosophila melanogaster*.
(TIF)

**S4 Fig. Cep162 is required for BBs docking to the plasma membrane in spermatocytes.** (A) Sequence confirmation of the deletion in *cep162* mutant. Primers used for sequence are marked with orange frames. The locations of 2 gRNAs used for mutant generation are underlined with black. Red and blue frames label the boundary of deletion *cep162* in mutant. (B) Genotyping of *cep162* flies using PCR. The amplification products were 881 bp long for WT and 555 bp long for mutants. (C) Immunostaining of CG6652 in spermatocyte cilia of WT or

*cep162¹* testis. CG6652 (green) marks the ciliary axoneme. In *cep162¹*, a few centrioles have over elongated CG6652 signals (arrows). (D) Live imaging of the connection between the ciliary cap and the plasma membrane in WT flies and *cep162* mutants. The plasma membrane (PM) was labeled with CellMask, and the BBs were marked by UNC-GFP. Defective connection between the BBs and the membrane was observed in some spermatids of *cep131¹* mutant (arrows). (E) Spermatocytes showing abnormal acetylated-tubulin extensions in centriole/BBs in *cep162; cep290¹* mutants. Scale bars, 5 μm (C, E), 10 μm (D).
(TIF)

**S5 Fig. Synthetic defects in Cep290 localization in *cep162; fam92* or *fam92; cep290^{ΔC}* mutants.** Cep290 are absent from the TZ in the spermatocyte of *cep162; fam92 or fam92; cep290^{ΔC}* mutants. The BB was labeled with γ-Tubulin (red). The error bars represent the mean ± SD, *n* = 30. Scale bars, 2 μm. The data underlying this figure can be found in S1 Data.
(TIF)

**S1 Raw Image. Full scans of western blots.**
(TIF)

**S1 Data. Raw data underlies the figures.**
(XLSX)

## Acknowledgments

We thank Dr. Wei Zhang from Tsinghua University for strains. We thank core facilities of *Drosophila* Resource and Technology (SIBCB, CAS), confocal imaging core facilities (SIAT, CAS), and EM core facilities (SIPPE, CAS) for their technical support.

## Author Contributions

**Conceptualization:** Zhimao Wu, Yingying Zhang, Bénédicte Durand, Qing Wei.

**Data curation:** Zhimao Wu, Huicheng Chen, Yingying Zhang, Qing Wei.

**Formal analysis:** Zhimao Wu, Yingying Zhang, Qiaoling Wang, Qing Wei.

**Investigation:** Zhimao Wu, Yingying Zhang.

**Methodology:** Zhimao Wu, Huicheng Chen, Yaru Wang, Qiaoling Wang, Yanan Hou.

**Project administration:** Zhimao Wu, Yingying Zhang, Yuejun Fu, Qing Wei.

**Resources:** Zhimao Wu.

**Software:** Zhimao Wu, Yingying Zhang.

**Supervision:** Zhimao Wu, Yingying Zhang, Qiaoling Wang, Bénédicte Durand, Qing Wei.

**Validation:** Zhimao Wu, Huicheng Chen, Yingying Zhang, Yaru Wang, Qiaoling Wang, Céline Augière, Yanan Hou.

**Writing – original draft:** Zhimao Wu, Huicheng Chen, Yingying Zhang, Céline Augière, Yuejun Fu, Ying Peng, Bénédicte Durand, Qing Wei.

**Writing – review & editing:** Zhimao Wu, Ying Peng, Bénédicte Durand, Qing Wei.

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
