## [Editor Report · Decision Letter 0]

6 Sep 2023

Dear Dr Wei, 

Thank you for submitting your manuscript entitled "Cep290 is cooperatively maintained at basal body by Cep131-Cep162 and Cby-Fam92 modules" for consideration as a Research Article by PLOS Biology.

Your manuscript has now been evaluated by the PLOS Biology editorial staff as well as by an academic editor with relevant expertise and I am writing to let you know that we would like to send your submission out for external peer review. However, we would like to consider it as an Update Article, so please select that article type from the drop down menu when you submit the metadata (see below).

Before we can send your manuscript to reviewers, we need you to complete your submission by providing the metadata that is required for full assessment. To this end, please login to Editorial Manager where you will find the paper in the 'Submissions Needing Revisions' folder on your homepage. Please click 'Revise Submission' from the Action Links and complete all additional questions in the submission questionnaire.

Once your full submission is complete, your paper will undergo a series of checks in preparation for peer review. After your manuscript has passed the checks it will be sent out for review. To provide the metadata for your submission, please Login to Editorial Manager (https://www.editorialmanager.com/pbiology) within two working days, i.e. by Sep 08 2023 11:59PM.

Kind regards,

Ines

--

Ines Alvarez-Garcia, PhD

Senior Editor

PLOS Biology

---

## [Decision Letter · Decision Letter 1]

12 Oct 2023

Dear Dr Wei,

Thank you for your patience while your manuscript entitled "Cep290 is cooperatively maintained at basal body by Cep131-Cep162 and Cby-Fam92 modules" went through peer-review at PLOS Biology as an Update Article. Your manuscript has now been evaluated by the PLOS Biology editors, an Academic Editor with relevant expertise, and by two independent reviewers.

The reviews are attached below. You will see that the reviewers find the conclusions interesting and recommend pursuing the paper for publication after addressing several issues. Reviewer 1 would like you to clarify several points regarding your previous paper, such as the relationship between Cep290 and the Cby-Fam92 module during ciliogenesis, among other issues. Reviewer 2 only raises several minor points that should be easy to address.

In light of the reviews, we are pleased to offer you the opportunity to address the comments from the reviewers in a revision that we anticipate should not take you very long. We will then assess your revised manuscript and your response to the reviewers' comments with our Academic Editor aiming to avoid further rounds of peer-review, although might need to consult with the reviewers, depending on the nature of the revisions.

**IMPORTANT - SUBMITTING YOUR REVISION**

3. Resubmission Checklist

a) *PLOS Data Policy*

b) *Published Peer Review*

Sincerely,

Ines

--

Ines Alvarez-Garcia, PhD

Senior Editor

PLOS Biology

Reviewers' comments

Rev. 1:

This paper by Wu et al. discusses the regulation of CEP290 localization during ciliogenesis in Drosophila. The authors demonstrate that the Cep131-Cep162 module near the axoneme and the Cby-Fam92 module at the membrane cooperate to control the basal body localization of Cep290, facilitating transition zone assembly during the initiation of ciliogenesis. Previous research (Vieillard et al., 2016) has already shown that Cep131 and Cby collaborate to recruit Cep290 and build the transition zone. However, this study adds a novel contribution by highlighting the essential role of Cep162, which bridges Cep131 and the C-terminus of Cep290 in transition zone formation. Additionally, the authors provide evidence that Dzip1-Cby-Fam92 interacts with the N-terminus of Cep290, playing a role in Cep290 recruitment. In summary, the paper proposes a model in which three central molecular pathway modules—Cep131-Cep162, Cep290, and Dzip1-Cby-Fam92—cooperatively regulate the initiation of cilium assembly. This work contributes to our understanding of the critical role of Cep290 in ciliogenesis and its regulation by other ciliogenesis modules, shedding light on related ciliopathies. I do have some questions/concerns that should be addressed.

Major Points:

1. In their previous paper, the authors showed that Cep290 acts upstream of Dzip1 and Cby (Wu et al., 2020. While Dzip1 depletion abolishes Cby localization at the TZ, but the recruitment of Cep290 was not affected. In this study, they conclude that the Dzip1-Cby-Fam92 module controls the basal body localization of Cep290. Thus, the relationship between Cep290 and the Cby-Fam92 module during ciliogenesis requires further clarification.

2. Previous work conducted in human cells has demonstrated that the localization of CEP290 at centriole distal ends is independent of CEP162, whereas CEP162 primarily recruits CEP290 to microtubules rather than basal bodies (Wang et al., 2013). Here you report a significant reduction in Cep290 signal at the basal body with mutated Cep162 in Drosophila (Fig. 3E). How can you explain this difference between human and fly?

3. Could the quantification of the levels of GFP-fused proteins at the basal body be explained by differences in total expression levels of these GFP-fused proteins in wild-type and mutated Drosophila lines (Fig 1A, 1B, 2D, 3E, 4A, etc.). It's important to ensure that the reduction of these proteins in is not caused by protein expression.

4. The dynamic and temporal localization pattern of Cep131 and Cep162 during cilia development is interesting (Fig. 2E). Given that these two proteins also impact the levels of Cep290 on the basal body during the initiation of ciliogenesis, it raises the question of whether the levels of Cep290 at the cilia change during different stages of spermatocyte cilia formation. Evaluating this by quantifying the intensity of Cep290 at different stages of ciliogenesis in spermatocyte could provide more insight into this mechanism.

5. Given the differences in findings between human and fly it is important to be more specific about conclusions being made in the text. For example, description of findings to in the "Abstract".

Rev. 2:

CEP290 is a major structural element of the ciliary transition zone that was shown initially in Chlamydomonas to link the axoneme to the ciliary membrane. It has a central role in gating ciliary transport. Human CEP290 mutations are associated with a range of ciliopathy syndromes including but limited to MKS, JBTS, BBS, LCA, and SLS. This work builds on previous work showing that Cep290 integration into the transition zone requires Cep131 and Cby, and on prior work indicating that Cep290 associates with the axoneme through the C-terminal domain, and with the ciliary membrane through the N-terminal domain.

The authors provide strong evidence in Drosophila that Cep290 is integrated into the transition zone through association by its C-terminal region with a Cep131-Cep162 module at the axoneme, and with the ciliary membrane by its N-terminal region by association with the Cby-Fam92 module. They examine this molecular organization in spermatocyte cilia/spermatid flagella and in sensory cilia.

The authors show that Cep131 and Cep162 interact directly and generated deletion alleles of both genes. They additionally showed that Cep162 associates with Cep290 at its C-terminal region, providing a link between Cep290 and its association with the axoneme. They further show that the N-terminal region of Cep290 associates with the Cby-Fam92 module at the ciliary membrane. Mutation of either Cep131 or Cep162, together with either Cby or Fam92 produces a severe loss of cilia similar to the phenotype of loss of Cep290.

Overall, the work advances our understanding of how Cep290 integrates into the transition zone of cilia. A model depicting an epistatic ordering of the assembly of the components is supported by the findings and presented in Figure 7. The work is clearly communicated, and the experiments are well-presented and support the major findings of the work.

I have some minor points to address:

1. For the deletion mutations generated for the cep131[1] mutant and the cep162[1] mutant, cep131[1] is described as a frameshift mutation in Supp Fig 1, and the same for cep162[1] in the legend for Figure 3. Please include the details in the Supp Figure 4. The authors should report what the predicted protein products are. The authors have the sequence data to discern this. For both mutants use standard nomenclature (eg "p.Arg131ProfsTer12"). This information could be included in Supp Figure 1 for cep131, and Supp Figure 4 for cep162.

2. The assembly of the long spermatid flagellum is unusual in that it is assembled in the cytoplasm while only the distal tip is encased in ciliary membrane, a ciliary cap, with a 'ring centriole' at its base. The authors show that in spermatocytes that are deficient in anchoring of centrioles to the membrane, the axoneme appears to overgrow. This is sometimes referred to in the manuscript as an axoneme and other times as 'abnormal extensions' that label with CG6655-GFP (Figures 5,6,S1,S4,), but in other contexts with Cep162, or Cep290 (Figure 4), indicating that they are something other than axonemes and maybe are also TZ-like. There are multiple figures where CG6655-GFP is used to label these axonemes/abnormal extensions as an apparent indication that the centriole did not anchor to the membrane. The authors should clarify what these 'abnormal extensions of the ciliary axonemes' are, or describe what they perceive them to be. Moreover, what is the consequence in spermatids - do the axonemes grow despite lacking a ciliary cap?

3. In Figure 3D, TEM of the cep162 mutant is presented and shown to have fewer numbers of spermatids. Several of the spermatids are colorized but this is not described in the text or figure legend. The colorized ones appear to have very small mitochondrial derivatives, and it appears that overall the sizes of the mitochondrial derivatives are variable compared to WT. This feature of the mutant should be included in the analysis and discussed in the context of other publications describing links between cilium proteins and mitochondrial morphogenesis in Drosophila testis.

4. The legend for Figure 3 refers to a panel E with images showing the pattern of CG6652-GFP in the cep162 mutant, but this panel is missing. Panel E is described as F in the legend. The text also does not refer to these data or a panel F, so perhaps the legend info on the CG6652-GFP localization in the cep162 mutant spermatocytes should be deleted.

5. In Figure 4A, the cep162[1] cep290[1] double mutant is depicted in the graphic as having extended filaments, probably TZ/axonemes, presumably because they infer that the mutant centrioles do not anchor to the membrane, but no evidence is shown for their assembly. The authors can check whether CG6652-GFP labels distal filaments in the double mutant, or perhaps staining for acetylated tubulin will suffice to show these extensions. Also, in the legend for panel D, referring to ciliary signal for Cep290-C, the statement "Notably, Cep290-C::GFP signal is completely lost in cep162; cep2901 double mutants." should instead say 'significantly reduced'.

6. Figure 5: The absence of axonemes in the cep162; cby TEM images is difficult to interpret as there are no distinguishing features in the image (like mitochondrial derivatives). It might be more straightforward and informative to characterize the extent of axoneme assembly in these mutant spermatids by staining for acetylated tubulin (as was done in Figure 2E). It is worth deeper inspection because others have shown that spermatids can elongate even without centrioles (eg sak mutant spermatids), as elongation is driven by the mitochondrial derivative and does not require the axoneme. Perhaps these mutants are unexpectedly more severe than mutants lacking centrioles.

7. In the Discussion, the authors propose that a ciliopathy phenotype might arise from double mutations in CEP131 or CEP162 with CBY1 or CIBAR1/2 (FAM92A/B). I assume part of this logic arises because no ciliopathies have been yet found associated with mutations in these genes, which is a point that should be stated. It is a worthwhile prediction, but it should just be pointed out that these genes are not yet clinically linked, while CEP290 is, and is likely attributed to the cooperative nature of these modules.

Additional points:

Manuscript needs page numbers.

Please run a spellcheck on the ms, there are numerous typos.

Typos in Introduction: "strucutres", "Chibbly", "foramtion", "seemly", "cordinating", "stablize". In Results: "Collectivley". In Discussion: "accessbile". In legends: "Trpplates"

---

## [Editor Report · Decision Letter 2]

11 Dec 2023

Dear Dr Wei,

Thank you for your patience while we considered your revised manuscript entitled "Cep290 is cooperatively maintained at basal body by Cep131-Cep162 and Cby-Fam92 modules" for publication as a Update Article at PLOS Biology. This revised version of your manuscript has been evaluated by the PLOS Biology editors and the Academic Editor.

Based on our Academic Editor's assessment of your revision, we are likely to accept this manuscript for publication, provided you address the data and other policy-related requests stated below.

In addition, we would like you to consider a suggestion to improve the title:

"Cep131-Cep162 and Cby-Fam92 complexes cooperatively maintain Cep290 at the basal body and contribute to ciliogenesis initiation"

We expect to receive your revised manuscript within two weeks. 

*Published Peer Review History*

*Press*

Sincerely,

Ines

--

Ines Alvarez-Garcia, PhD

Senior Editor

PLOS Biology

Fig. 1A, B; Fig. 2C, D, G, H; Fig. 3B, C, E; Fig. 4A, D; Fig. 5A, C, E, F, Fig. 6A, C-F; Fig. S1D, E and Fig. S5

---

## [Editor Report · Decision Letter 3]

31 Jan 2024

Dear Dr Wei,

Thank you for the submission of your revised Update Article entitled "Cep131-Cep162 and Cby-Fam92 complexes cooperatively maintain Cep290 at the basal body and contribute to ciliogenesis initiation" for publication in PLOS Biology. On behalf of my colleagues and the Academic Editor, Renata Basto, I am delighted to let you know that we can in principle accept your manuscript for publication, provided you address any remaining formatting and reporting issues. These will be detailed in an email you should receive within 2-3 business days from our colleagues in the journal operations team; no action is required from you until then. Please note that we will not be able to formally accept your manuscript and schedule it for publication until you have completed any requested changes.

PRESS

Sincerely, 

Ines

--

Ines Alvarez-Garcia, PhD

Senior Editor

PLOS Biology
